# A pistil-expressed pectin methylesterase confers cross-incompatibility between strains of *Zea mays*

Yongxian Lu [1], Samuel A. Hokin[1], Jerry L. Kermicle[2], Thomas Hartwig[1] & Mathew M.S. Evans [1]

A central problem in speciation is the origin and mechanisms of reproductive barriers that block gene flow between sympatric populations. Wind-pollinated plant species that flower in synchrony with one another rely on post-pollination interactions to maintain reproductive isolation. In some locations in Mexico, sympatric populations of domesticated maize and annual teosinte grow in intimate associate and flower synchronously, but rarely produce hybrids. This trait is typically conferred by a single haplotype, *Teosinte crossing barrier1-s*. Here, we show that the *Teosinte crossing barrier1-s* haplotype contains a pistil-expressed, potential speciation gene, encoding a pectin methylesterase homolog. The modification of the pollen tube cell wall by the pistil, then, is likely a key mechanism for pollen rejection in *Zea* and may represent a general mechanism for reproductive isolation in grasses.

[1] Department of Plant Biology, Carnegie Institution for Science, 260 Panama St, Stanford, CA 94305, USA. [2] Laboratory of Genetics, University of Wisconsin, 425 Henry Mall, Madison, WI 53706, USA. Correspondence and requests for materials should be addressed to M.M.S.E. (email: mevans@carnegiescience.edu)

Reproductive isolation is essential for the origin of new species[1]. Wind-pollinated plant species, like those of the grass family, rely on post-pollination barriers to prevent hybrid production, including both pollen–pistil interactions that prevent fertilization and post-fertilization hybrid abortion or sterility[2–4]. Maize (*Zea mays* ssp. *mays*) was domesticated from annual teosinte (*Z. mays* ssp. *parviglumis*) in the Balsas River valley of Mexico about 9000 years ago[5]. Despite this recent divergence and the full compatibility of some maize and teosinte populations, other sympatric populations of annual teosinte grow together and flower synchronously with domesticated maize, but rarely produce hybrids[6,7]. The *Teosinte crossing barrier1-s* (*Tcb1-s*) haplotype, originally identified from strains of *Z. mays* ssp. *mexicana* teosinte, confers this cross-incompatibility trait to these plants.

In *Z. mays*, haplotypes at three loci, *Tcb1-s*, *Gametophyte factor1-s* (*Ga1-s*) and *Gametophyte factor2-s* (*Ga2-s*) confer unilateral cross-incompatibility against varieties carrying the *tcb1* (or *ga1* or *ga2*, respectively) haplotype. While *Ga1-s* and *Ga2-s* are widespread in domesticated maize, *Tcb1-s* is almost exclusively found in wild teosinte populations, with the exception of one ancient Maiz Dulce sweet corn variety from Mexico[8]. *Tcb1-s* was first described in teosinte subspecies *mexicana* Collection 48703 from central and southern Mexico[6,7,9]. Other collections of teosinte of both *mexicana* and *parviglumis* subspecies from the central Mexican plateau also carry *Tcb1-s*[10]. Since *Tcb1-s* is unilateral it can be crossed into maize and functions as it does in teosinte to reject standard maize pollen[7].

*Tcb1-s* females block fertilization by maize (*tcb1* type) pollen by restricting pollen tube growth in the pistil (silk)[4]. This function makes *Tcb1* a candidate speciation gene contributing to isolation of diverging maize and teosinte populations, as wild teosinte populations respond to the pressure of cultivated, closely related varieties of domesticated maize[10]. In the reciprocal cross, teosinte pollen is able to fertilize maize, although at a slight disadvantage when in competition with maize pollen[7]. The *Tcb1-s* locus contains male (*Tcb1-male*) and female (*Tcb1-female*) genes, which are tightly linked but separable by recombination[4]. Thus, there are four functional haplotypes at this locus (Supplementary Table 1 for gene content and origin): *Tcb1-s* has both functional male and female genes, *Tcb1-m* has only the functional male gene[4,10], *Tcb1-f* has only the functional female gene, and the *tcb1* haplotype found in almost all maize lines has neither of the two functional genes. In teosinte, *Tcb1-f* activity in the silks prevents fertilization by maize (*tcb1*) pollen, while *Tcb1-m* activity in pollen enables fertilization of *Tcb1-f* or *Tcb1-s* females[4].

There are two alternative explanations of rejection of *tcb1* pollen: (1) *tcb1* pollen expresses an allele of *Tcb1-m* that is recognized by the *Tcb1-female* gene leading to directed inhibition of *tcb1* but not the unrecognized *Tcb1-m* pollen; or (2) *Tcb1-m* confers a function to pollen not present in *tcb1* allowing it to overcome the barrier set up indiscriminately in *Tcb1-f* pistils. This second case is preferred since pollen heterozygous for *Tcb1-s* and *tcb1* haplotypes (using a trisomic line carrying a duplication of the short arm of Chromosome 4) fertilizes *Tcb1-s* females, indicating that the *Tcb1-female:Tcb1-male* match overcomes the *Tcb1-female:tcb1-male* mismatch (the situation is similar for the *Ga1* and *Ga2* systems)[11,12].

This study demonstrates that the *Tcb1-female* gene encodes a pistil-expressed pectin methylesterase38 (PME38) homolog. The identity of this gene suggests a mechanism for reproductive isolation in diverging plant populations of maize and some teosintes, in which the pollen tube cell wall is modified by the female thus preventing continued pollen tube growth and delivery of the sperm cells. Agriculturally, this work may facilitate reproductive isolation of specialty crops, and

enrichment of crop plant germplasm by overcoming barriers to crossing with wild relatives[13].

## Results

**Fine mapping analysis and mutant screen of *Tcb1-s*.** To clone the *Tcb1* genes, fine mapping of *Tcb1-s::Col48703* haplotype was performed based on a *tcb1* backcross population of ~15,000 chromosomes. Using the maize B73 genome as a reference[14], the *Tcb1* locus was delimited to a region spanning 480 kb on the short arm of Chromosome 4. Within this region, there are 10 annotated genes. However, all of these were ruled out as candidates for *Tcb1* functions because they either had identical sequence with identical expression levels between *tcb1* and *Tcb1-s* haplotypes or no expression in the silk or pollen in *Tcb1-s* or *tcb1* (mapping markers included in Supplementary Table 2). The *Tcb1* genes, therefore, are likely absent from the maize genome. This is not surprising considering the widespread structural variations in genomes between maize lines and between teosinte populations[15]. Using gene content information from B73, clones were isolated and sequenced from a Bacteria Artificial Chromosome (BAC) library of the *Tcb1-s::Col48703* haplotype to identify gene models present in the *Tcb1-s* haplotype that are absent in maize reference genomes. These BACs were assembled into three contigs with two gaps in the *Tcb1-s* region. Four new gene models from this assembly could be eliminated as candidates based on lack of expression or identical sequence coupled with identical expression to unlinked maize genes. However, the data did provide more mapping markers for delineating the *Tcb1-f* and *Tcb1-m* gene regions.

To identify *Tcb1-f* knockout mutants, maize lines homozygous for the *Tcb1-s::Col48703* haplotype and carrying active *Mutator* transposons were crossed to maize inbred A195 *su1*. The progeny are expected to be heterozygous for *Tcb1-s* with *su1* ~6 cM away and in repulsion[7]. Due to the rejection of the *tcb1* pollen (which is predominantly *su1*), about 3% of the kernels in every ear with functional *Tcb1-f* were expected to be *su1* homozygotes in this open-pollinated population, while any ears without a crossing barrier that fail to exclude *tcb1* pollen were predicted to segregate *su1* at 25%. Out of a population of ~6000 individuals, two exceptional ears were found. One ear segregated for 25.6% sugary. This allele is termed *tcb1-f(KO1)*. The second isolate contained a sector of about 45 kernels within which the segregation was one-fourth sugary despite sugary segregating at ~3% over the rest of the ear. This allele is termed *tcb1-f(KO2)* and was recovered from the loss-of-function sector. Mixed pollination tests with the progeny of both individuals show that the loss of function is heritable, and both variants fertilized a *Tcb1-s/tcb1* strain normally, indicating the retention of the male function of *Tcb1-s* (*Tcb1-f* mutated, but *Tcb1-m* intact) (Fig. 1). In the case of *tcb1-f(KO2)*, progeny of seeds within the loss-of-function side of the ear inherited the knock-out, while those on the other side of the ear inherited fully functional *Tcb1-s*.

**Tcb1-female encodes a PME38 homolog.** RNA from silks of four genotypes were subjected to short read RNA-seq: the two knockout mutants, a standard maize inbred line W22 (genotype *tcb1*), and a functional *Tcb1-s* line (a W22 subline to which the *Tcb1-s::Col48703* haplotype had been introduced by back-crossing). Transcript models were assembled de novo from the RNA-seq reads, and expression levels of genes were compared between lines. One gene, encoding a maize PME38 homolog, was identified as a candidate for the *Tcb1-f* gene. This gene is highly expressed in *Tcb1-s* silks (with a peak read depth of ~100,000) compared to the standard maize *tcb1* W22 silks, *tcb1-f(KO2)* silks (maximum read depths of ~100) and *tcb1-f(KO1)* silks

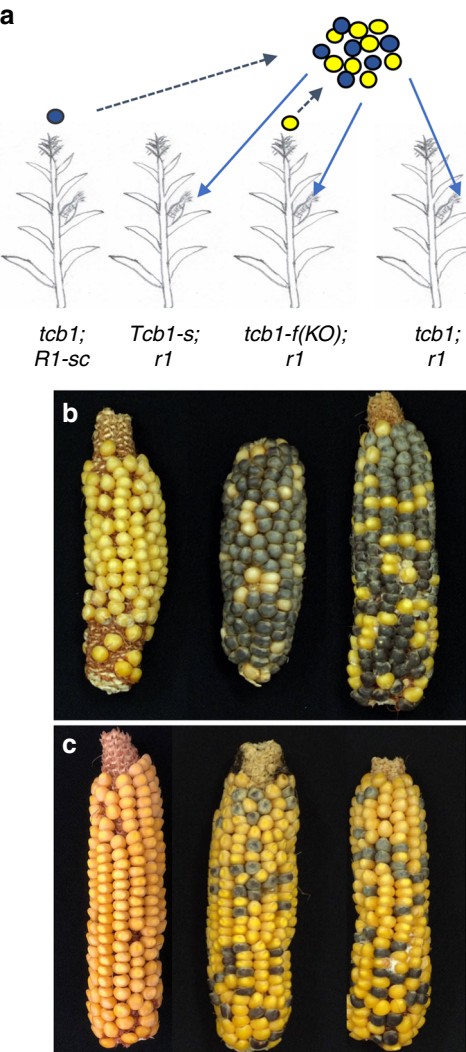

**Fig. 1** Mixed pollination test of the *Tcb1-f* mutants. **a** Scheme of the experiment: Two pollen donor lines and three pollen receiver lines were used. Pollen from a *tcb1; R1-self color* maize line (purple circles), produces purple kernels, while pollen from test plants (yellow circles) produces white or yellow kernels. Pollen from the two donors was mixed and put on three receiving ears: (1) *Tcb1-s* ears to verify the *Tcb1-male* function from the *KO* line; (2) *tcb1-f(KO)* ears to test the presence/absence of the female barrier in the knockout mutant individuals; and (3) a maize (*tcb1; r1*) neutral ear to identify the ratio of viable pollen grains from the two donors in the mixture. **b** Ears from the three pollen recipients for the *tcb1-f(KO1)*) test. **c** Ears from the three pollen recipients for the *tcb1-f(KO2)* test. In both tests, pollen from the *KO* plants successfully fertilized the *Tcb1-s* ear (left ear in **b** and **c**), while the ears from *KO* plants showed no barrier to *tcb1* maize pollen with a similar frequency of purple kernels on mutant ears and the neutral test ears (middle and right ears in **b** and **c**)

(maximum read depth of ~10,000 for the 5′ end and ~100 for the 3′ end of the transcript model) (Fig. 2a). qRT-PCR confirmed this expression difference (Fig. 2b). A BAC clone from the *Tcb1-s* library carrying this *PME38* homolog was identified by PCR. By comparing mRNA and BAC sequences, a 99-base pair intron was identified in *Tcb1-f* (Supplementary Fig. 1). Comparison of this gene to maize sequences (maizegdb.org and NCBI) by BLAST demonstrated that it is not present in the maize B73 reference genome, which is consistent with the mapping data. Its closest homologs are pseudogenes located at the *ga1* locus in both B73 and W22 and a silk-expressed (putative *Ga1-f*) gene in *Ga1-s*

maize[16]. It is likely that the small number of reads from *tcb1* RNA-Seq that map to the gene are a consequence of miss-assignment of reads from the poorly expressed pseudogenes.

In addition to the two knockout mutants above, several other lines derived from *Tcb1-s::Col48703* have lost female barrier function. One was recovered during early backcrossing of the *Tcb1-s::Col48703* haplotype into maize[6]. Mixed pollination confirmed this is a *Tcb1-male* only plant (Supplementary Fig. 2). Additionally, two independent *Tcb1-s* lines were isolated in which the barrier gradually lost strength over 10 generations of backcrossing to maize[4], and are named as *tcb1-f::silent lineage1* (*tcb1-f::sl1*) and *tcb1-f::silent lineage2* (*tcb1-f::sl2*), based on the progressive manner of the barrier loss. This *PME38* has much lower expression in these three lines than *Tcb1-s::Col48703* (Fig. 2b). Expression of the *PME38* was also tested on two *Tcb1-m* lines from the mapping population, which lost the *Tcb1-female* gene by recombination. Again, the *PME38* expression was much lower than in *Tcb1-s* lines (Fig. 2b).

A PCR-based derived Cleaved Amplified Polymorphic Sequence (dCAPS) marker was designed for the *PME38* gene (Supplementary Fig. 3). This marker was tested on the 15 closest recombinants from the mapping population of ~15,000 individuals (including four recombinants between the *Tcb1-f* and *Tcb1-m* genes)[4]. Of the 15 plants, 6 carried *Tcb1-f* and blocked maize pollen, and 9 lacked the barrier. Results showed that all the six recombinants that carry the barrier had the *PME38* gene, while in all nine recombinants that are receptive to maize pollen, this *PME38* was absent, demonstrating tight linkage between the *PME38* and *Tcb1-f* barrier function.

RNA-seq data suggest that the mutation in the *PME38* lies in the first exon in *tcb1-f(KO1)* (Fig. 2a). Quite differently, in *tcb1-f(KO2)* mutant silk RNA-seq reads had the same low level of expression as *tcb1* silks along the whole *PME38* transcript, and the whole coding of *PME38* can be PCR-amplified from *tcb1-f(KO2)* genomic DNA but not *tcb1-f(KO1)* (Supplementary Fig. 4). Whole genome resequencing of both mutants identified a *Hopscotch* retrotransposon insertion in the first exon in *tcb1-f(KO1)*, a mutation unrelated to *Mutator* transposons, but deletions and non-*Mutator* insertions occur in these lines[17]. This insertion is close to the site where the *PME38* expression drops sharply. PCR spanning both ends of the insertion confirmed the insertion event and the border sequences (Supplementary Fig. 5). In contrast, in *tcb1-f(KO2)*, the *PME38* gene was fully assembled, consistent with the PCR data. The *tcb1-f(KO2)* allele then could either be mutated in a regulatory region, potentially hundred kilobases away from the coding region, or could be an epi-allele. Similarly, no mutations were found in the coding region in the *Tcb1-m* line or the *tcb1-f::sl1* or *tcb1-f::sl2* lines described above.

**Reversion of silenced *tcb1-f* loss-of-funtion lines.** Since several of the loss-of-function lines did not carry mutations in the *PME38* coding region, tests were performed to determine if they might be epi-alleles and so could revert back to full-strength *Tcb1-s*. The *tcb1-f(KO2)*, *tcb1-f::sl1*, and *tcb1-f::sl2* lines were tested for reversion to *Tcb1-s* in double mutants with the *mediator of paramutation1* (*mop1*) mutation. *MOP1* encodes a RNA-dependent RNA polymerase and is a key component of RNA-directed DNA methylation[18]. *mop1* mutations reactivate silenced genes and affect broad developmental programs[19]. Re-activation of the *Tcb1-f* function was rare. In only ~14–22% of the *mop1* females tested, did the loss-of-function plants show some recovery of *Tcb1-f* function. Pollen competition experiments were performed for full strength, wild-type *Tcb1-s* females, *tcb1* females, and the *tcb1-f* loss of function lines without sequence changes (*tcb1-f(KO2)*, *tcb1-f::sl1*, *tcb1-f::sl2*, and a few *Tcb1-m*)

(Fig. 3 and Source Data file). All of the *Tcb1-s* ears tested showed strong preference for *Tcb1-s* pollen (0–7% kernels from *tcb1* pollen regardless of the ratio of the two pollen types in the mix as indicated by the neutral ear) with the kernel ratio on the test ear and control ear being different from each other at $p < 0.0001$ (two-tailed Fisher exact test) (Fig. 3b). Of the 36 *mop1; tcb1-f* loss of function females tested only one (*tcb1-f::sl1; mop1*) had as strong of a pollen preference as full strength *Tcb1-s* females, but five had a difference between the test and control ears at $p < 0.0001$ and an additional three females could be included if the stringency was relaxed to $p < 0.01$ (two-tailed Fisher exact test) (Fig. 3b). These partial revertants included plants of lines *tcb1-f (KO2)*, *tcb1-f::sl1*, and *tcb1-f::sl2*. Of the 12 loss of function plants tested that were heterozygous wild-type for *mop1*, none of the

plants passed the more stringent $p < 0.0001$ threshold and one passed the less stringent $p < 0.01$ threshold (two-tailed Fisher exact test). Additionally, none of the 13 comparisons between two *tcb1* ears had significant differences between the two ears.

A subset of homozygous *mop1 tcb1-f::sl2* plants were tested at random for *PME38* expression in silks prior to pollination. Among the seven tested plants, one plant, yx57-13, showed increased expression, about 400-fold higher expression compared to that of the standard W22 maize and eight times higher than *tcb1-f::sl2* plants (Fig. 3c). This plant was the only one of the seven tested for expression that recovered the ability to reject *tcb1* pollen, although not as efficiently as full strength *Tcb1-s* plants, which have still higher expression of the *PME38* gene. This indicates a correlation between *PME38* expression level and the

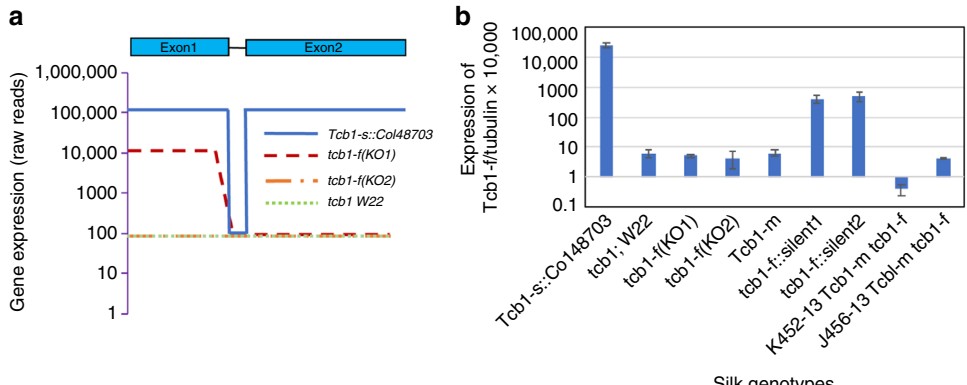

**Fig. 2** Gene expression profiling identifies a *PME38* as a *Tcb1-f* candidate gene. RNA samples collected from silks of different genotypes was analyzed by RNA-Seq and RT-PCR. **a** *Tcb1-f/PME38* gene structure is shown above the graph (solid line indicates single intron), and RNA-Seq read depth is shown for the *Tcb1-f/PME38* gene. **b** *Tcb1-f/PME38* gene expression, compared to *tubulin* levels as a control, as measured by qRT-PCR in *Tcb1-s* and loss-of-function *Tcb1-s* lines: *Tcb1-s*, full strength *Tcb1-s* barrier line; *tcb1-f(KO1)* and *tcb1-f(KO2)*, two loss-of-function alleles from a *Mutator* transposon mutagenesis; *Tcb1-m*, a spontaneous *Tcb1-male* only line; *tcb1-f::silent lineage1* and *tcb1-f::silent lineage2*; and *K452-13* and *J456-13*, two *Tcb1-male* only lines that lost *Tcb1-f* by recombination. Error bars equal the standard error of the mean. Source data for RT-PCR are provided as a Source Data file. RNA-seq reads provided in NCBI BioProject PRJNA528983

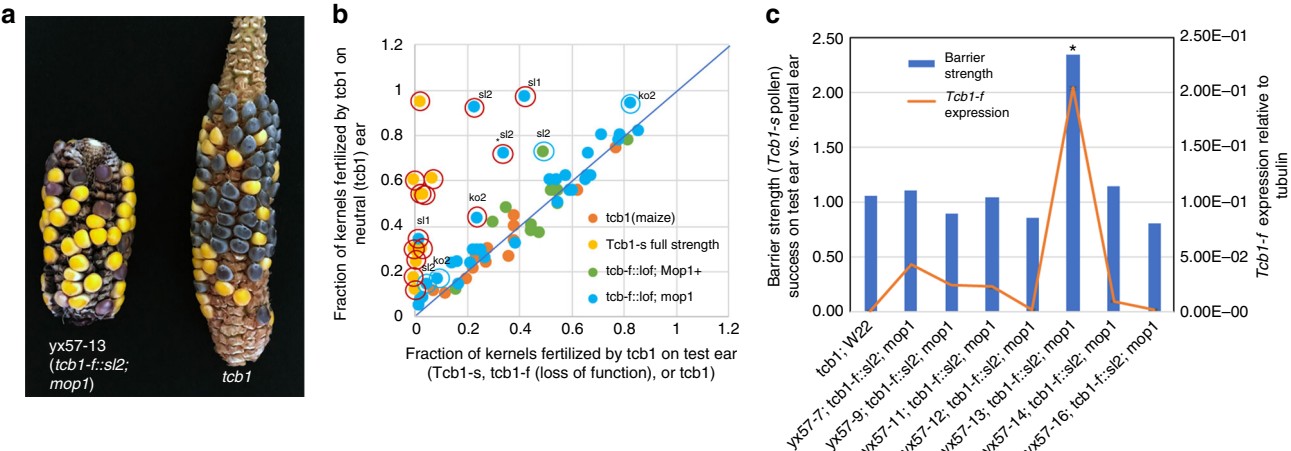

**Fig. 3** Reversion of *tcb1-f* loss-of-function. **a** Revertant ear (marked by asterisk in **b** and **c**) pollinated by a mix of *tcb1; R1-sc* and *Tcb1-m; r1* pollen showing higher frequency of yellow kernels on the test ear than the *tcb1* control ear. **b** Results of mixed pollination tests on four genotypes: *tcb1*, *Tcb1-s*, *tcb1-f(loss-of-function(lof))* alleles, and *tcb1-f(lof) mop1* double mutants. The percentage of kernels from *tcb1* pollen on the test ear are plotted on the x-axis and the percentage of kernels from *tcb1* pollen on the control *tcb1* ear on the y-axis. Equal percentages in the two ears indicates no barrier (line with slope = 1). Red circles indicate ears with significantly fewer *tcb1* kernels in the test ear vs. the control ear at $p < 0.0001$ and blue circles at $0.0001 < p < 0.01$ (two-tailed Fisher exact test). The loss-of-function line is indicated for each revertant. **c** *Tcb1-f/PME38* gene expression level and barrier strength. Barrier strength is expressed as the ratio of kernels from *Tcb1-s* vs. *tcb1* pollen on the test ear vs. the control *tcb1* ear (columns), with a fraction of 1.0 indicating no barrier and significantly higher values a functional barrier. *Tcb1-f/PME38* RNA levels are expressed as relative to the *tubulin* control gene (orange line). Error bars in **c** equal the standard error of the mean. Individual pedigree numbers are given for loss-of-function plants tested for reversion in **a** and **c**. Source data for panels **b** and **c** is provided in a Source Data file

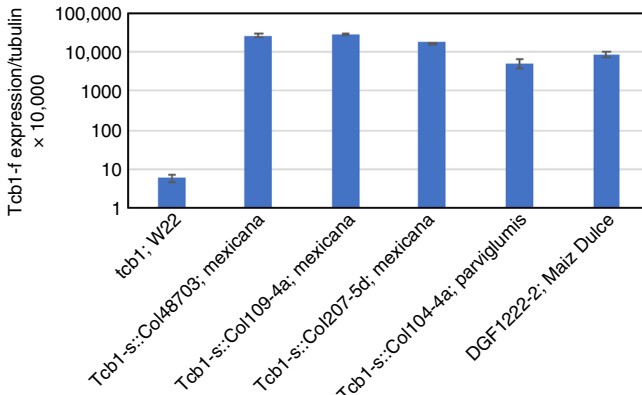

**Fig. 4** Mean expression of *Tcb1-f/PME38* in lines with different *Tcb1-s* haplotypes. W22 is a standard maize *tcb1* line, *Tcb1-s::Col48703*, *Tcb1-s:: Col109-4a*, and *Tcb1-s::Col207-5d* are three independent collections of *Zea mays* ssp. *mexicana* teosinte lines; *Tcb1-s::Col104-4a*, a *Zea mays* ssp. *parviglumis* line; and DGF1222-2 derived from Maiz Dulce, an ancient Mexican maize sweet corn variety[8]. Error bars equal the standard error of the mean. Source data for RT-PCR are provided as a Source Data file

female barrier strength with the expression level in *tcb1-f::sl1* and *tcb1-f::sl2* being below the expression threshold for producing a detectable barrier. This correlation and the *Hopscotch* insertion in *tcb1-f(KO1)* supports this *PME38* as the *Tcb1-f* gene.

In addition to the *Tcb1-s::Col48703* strain descried above, three other teosinte-derived *Tcb1-s* lines, two from ssp. *mexicana* and one ssp. *parviglumis*[10], were tested for *Tcb1-f/PME38* expression in silk tissue. In all three lines, *Tcb1-f/PME38* expression levels are extremely high and comparable to that of the original central plateau *TIC* haplotype *Tcb1-s::Col48703* (Fig. 4). Interestingly, even though none of the modern north American maize lines tested to date carry the *Tcb1-s* haplotype, *Tcb1*-s was identified in lines descended from an ancient Maiz Dulce variety, Jalisco78 that grows at intermediate altitudes in southwestern Mexico, during a survey for cross-incompatibility factors[8]. This is a specialty line that may have undergone selection for cross-incompatibility factors similarly to *Ga1-s* in maize popcorn lines but in this case to maintain a sweet corn trait. Whether this Maiz Dulce line acquired *Tcb1-s* from nearby teosinte populations during its origin is unknown, but maize lines from this region have been shown to have substantial introgression from ssp. *mexicana* teosintes[20].

**Diversity and relationship of *Tcb1-f* to other *PME* genes**. Predicted *Tcb1-f/PME38* coding sequences are identical in all five *Tcb1-s* lines: three *mexicana* accessions, one *parviglumis* accession, and the Maiz Dulce line. One single nucleotide polymorphism (SNP) in the intron separates these lines into two groups: one group including the *parviglumis* line (Col104-4a) and one *mexicana* line (Col109-4a), and the other group including two *mexicana* lines (Col48703 and Col207-5d) and the Maiz Dulce line (Supplementary Fig. 6).

The most similar gene to *Tcb1-f/PME38* is a candidate PME gene for *Ga1-female* function. This gene, termed *ZmPME3*, was found to be expressed in the silks of *Ga1-s*, but not in *ga1* silks, and maps to the *Ga1* locus[16]. Alignment of the *ZmPME3* and *Tcb1-f/PME38* show that the two PMEs differ in eight amino acids (Supplementary Fig. 7). The number of polymorphisms (15 of 1296 nucleotides) between *Tcb1-f/PME38* and *ZmPME3* suggests that these two genes diverged ~175,000 years ago, well before the split between the *mexicana* and *parviglumis* subspecies of teosinte and just before the split between *Z. mays* and *Zea*

*luxurians*, using calculated nucleotide substitution rates for maize[21] and a calculated time since the split between *mexicana* and *parviglumis* of ~60,000 years and *parviglumis* and *luxurians* of ~140,000 years[22]. It will be interesting to test whether the *Tcb1-male* and *Ga1-male* genes diverged at a similar time, suggesting that the male–female genes were paired before divergence of *Tcb1-s* and *Ga1-s*.

## Discussion

*Tcb1-s* and *Ga1-s* are mostly cross-incompatible with one another, suggesting the male genes are also divergent. However, *Tcb1-s* and *Ga1-s* are not fully cross-incompatible. In situations where pollen rejection is not absolute, *Tcb1-s* pollen has a competitive advantage over *tcb1* pollen on *Ga1-s* or *Ga2-s* silks. This is true for all combinations of interactions between crossing barrier loci[7,12] and is consistent with them encoding related proteins. The confounding result that the behavior of pollen tubes during rejection by each system is slightly different may be explained by differences in amino acid sequence or differences in expression level or pattern between the haplotypes[4]. *Tcb1-f/PME38* encodes a group 1 type of PME without an N-terminal pectin methylesterase inhibitor domain[23], and contains a predicted signal peptide, so it has the potential to be secreted and interact directly with the pollen tube to remove methyl-esters from the pectin wall of the pollen tube. Esterified pectins are typically associated with the tip of the growing pollen tube, while de-esterified pectins are enriched distally, and there is a correlation between pectin de-esterification and increased cell wall stiffness[24]. Pollen cells finely tune the stiffness of the tip cell wall to sustain pollen tube elongation. Either under-supply or over-supply of PME activity can result in disturbed pollen tube growth and compromised male fertility[25–28]. The TCB1-F (and ZMPME3) protein falls into the Plant 1a clade of mature PME enzymes[29] (Supplementary Fig. 8).

In summary, genetic and genomic data identify the *Tcb1-female* barrier gene as a *PME38* homolog. Teosinte lines carrying *Tcb1-f* block maize pollen that lacks the male function provided by *Tcb1-m*. That the *Tcb1-f* gene encodes a cell wall modifying enzyme is consistent with the result that heterozygous *Tcb1-s/tcb1* pollen is functional on *Tcb1-s* silks rather than active targeting of a pollen protein encoded by (a hypothetical maize allele of) the *tcb1-m* gene[11]. Surprisingly, it was shown that another PME family member is encoded by the *Ga1-male* gene[30] (in a very distinct clade, Plant X2, of PME enzymes (Supplementary Fig. 8)), raising the possibility that the biochemical barrier to pollen and the ability of pollen to overcome that barrier are conferred by different classes of PME proteins. It will be interesting to test how universal this barrier mechanism is in sexual plant reproduction. The grasses also have an unusually high species diversity for a family with abiotic pollinators[31]. Identification of the *Tcb1-female* gene may facilitate research into the mechanisms of speciation in the grasses.

## Methods

**Maize and teosinte lines and growth conditions**. All *Tcb1-s* lines used in this study have had the *Tcb1-s* haplotype introgressed from their maize or teosinte line of origin into standard maize lines W22 or B73 by backcrossing. The location of origin of these lines has been described elsewhere[4,7,8,10]. Plants were grown under summer field conditions at either Stanford, California or Madison, Wisconsin.

**Tcb1-s mapping**. As described before[7], a Central Plateau teosinte collection 48703[9] carrying the *Tcb1-s* barrier was backcrossed to the Mid-western US dent inbred W22 to incorporate the *Tcb1-s* locus into a maize background. This *Tcb1-s* strain was crossed to a chromosome 4 maize tester line *virescent17* (*v17*) *brown midrib3* (*bm3*) *sugary1* (*su1*), and the F1 was then backcrossed to the same tester line. Recombinants carrying crossovers between the four visual markers were tested for the *Tcb1-s* male and female functions in reciprocal crosses with *Tcb1-s/su1* F1 plants. PCR mapping markers were developed to refine the location of crossovers in these recombinants.

**Tcb1-f knockout mutant screen**. To identify loss-of-function mutants of *Tcb1-female*, a *Ga1-m Tcb1-s* active *Mutator* strain was crossed to maize inbred A195 *su1* (*tcb1*), and then the progeny were grown as an open-pollinated block. Most of the progeny are expected to be heterozygous for *Tcb1-s* and *su1* in repulsion with *su1* ~6 cM away from the *tcb1* locus[7]. Due to the rejection of the *tcb1* pollen (which is predominantly *su1*), about 3% of the kernels in every ear with functional *Tcb1-f* are expected to be *sugary* in this open-pollinated population, while those without a crossing barrier were predicted to segregate *su1* at 25%.

**Mixed pollination experiments**. For the mixed pollination testing of the two *tcb1-f* knockout mutants, two pollen donor lines and three pollen receiver lines were used. Pollen from a maize line (*tcb1*) that does not have the *Tcb1-s* barrier genes but carries the endosperm color marker *R1-self color* (*R1-sc*) will produce purple kernels after fertilization of the lines used, while pollen from the knockout plants and the *Tcb1-m* plants carry *r1-r* and produce anthocyaninless kernels that are white or yellow. After being collected from the two donors and mixed, pollen was put on the three receiving ears: (1) a *Tcb1-s* tester ear was used to verify the presence of the *Tcb1* male function from the *tcb1-f::KO* pollen; (2) the *tcb1-f (KO)* ear was used to test the presence/absence of the female barrier function in the knockout mutant; and (3) a maize (*tcb1*) neutral ear was used to assay the percentage of viable pollen grains from the two donors in the mixture. Because any two tassels cannot be counted on to produce the same amounts of pollen, the ratio of the two pollen types in any mix cannot be assumed to be 1:1. Consequently, it is essential to make a cross of each pollen mix onto a standard *tcb1* maize ear (with a known pollen acceptance/function bias of 60% *tcb1*: 40% *Tcb1-s*[10]) to verify the ratio of functional pollen grains in the mix. The same protocol was used on the spontaneous *Tcb1-m* plant, except the *Tcb1-m* plant being tested was substituted for the *KO* plant. For mixed pollinations of the *tcb1-f::silent lineage mop1* double-mutant plants, pollen from the same *R1-sc tcb1* line and a *r1-r Tcb1-s* tester line was collected, mixed, and applied to the individual silent line ears and the neutral maize ears.

**Silk tissue collection, RNA isolation, and cDNA synthesis**. Plants for RNA isolation were grown in summer field conditions in Stanford, CA. Silk tissues were collected around 11 a.m., immediately put into liquid nitrogen in the field, and stored at −80 °C. Total RNA was isolated from silks with Trizol reagent (Invitrogen), DNase-treated, and either subjected to Illumina short read paired end RNA-seq, or used to synthesize the first strand cDNA with the Superscript IV RT kit (Invitrogen).

**Quantitative RT-PCR**. RT-PCR was performed on a Roche Lightcycler 480II (Roche Diagnostics). Each line/genotype had three biological replicates, and each in turn had three technical replicates. For cases where expression levels in individual plants is shown three technical replicates were used. Tubulin (Zm00001d033850) was used as a reference gene. In each line, relative expression levels were obtained by comparing *Tcb1-f/PME38* to tubulin. Primers are listed in Supplementary Table 3.

**Sequencing, assembly, and analysis**. All the RNA and DNA sequencing works were done with Illumina Paired-end sequencing by Novogene (CA, USA). RNA-seq reads from all samples were combined and de novo assembled with Trinity v2.4.0[32]. The gene in contig DN33598_c7_g3_i1 was identified as the *Tcb1-f* candidate gene due to its extremely high expression in the functional *Tcb1-s* line and the near lack of expression in the *KO* mutants and a standard W22 maize line. PCR primers were designed based on the DN33598_c7_g3_i1 sequence, and one BAC clone was isolated from a library made from a maize line into which the *Tcb1-s::Col48703* haplotype had been introgressed. The BAC sequencing reads were assembled with SPAdes v3.11.1[33]. NODE_62, a contig that is 13,656 bp with coverage of 4029, was identified as having the *Tcb1-f* candidate gene. Whole genome sequencing reads from the two *KO* mutants were individually assembled with SPAdes v3.11.1 and BLASTed against NODE_62. Also, the mutant sequencing reads were mapped against NODE_62 using GSNAP[34]. Combining both approaches identified the *hopscotch* retrotransposon insertion in the *tcb1-f(KO1)*-mutant allele.

For phylogenetic analyses, alignments were made using the ClustalW algorithm in MegAlign (DNASTAR). The predicted mature PME enzymes and the Arabidopsis PME family members were taken from Markovic and Janecek, as were the subfamily designations[29]. Phylogenies were produced from these alignments using MrBayes v3.2.0 using default settings for amino acid analysis[35]. The MrBayes analysis was performed for 4,100,000 generations at which point the standard deviation of the split frequencies was below 0.004.

**Reporting summary**. Further information on research design is available in the Nature Research Reporting Summary linked to this article.

## Data availability

RNA-seq and genomic sequence data of Knockout lines and sequence reads of the BAC clone carrying *Tcb1-f* are available at NCBI BioProjects PRJNA528310 and

PRJNA528983 and PRJNA532627. Coding and genomic sequence of *Tcb1-f* are available at GenBank MK789594 and MK789593. Data supporting the findings of this work are available within the paper and its Supplementary Information files. The source data underlying Figs. 2, 3, and 4 are provided as a Source Data file. Seed stocks and other data are available from the authors.

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

## Acknowledgements

The authors would like to thank Kathy Barton for the encouragement, insight, and support that kept this project alive, and to thank members of the Evans and Barton labs past and present for helpful discussions. We would also like to thank Beverly Oashgar for help with the screen for loss-of-function mutants and David Heller, Lance Cabalona, Clayton Coker, Amber Glowacki, and Hannah Vahldick for help growing plants and making crosses, and Jeffrey Yen for help isolating BAC clones. We would also like to thank Jeffrey Ross-Ibarra for help in calculating the divergence time between *ZmPME3* and *Tcb1-f*. This work was supported by National Science Foundation Award number IOS-0951259 and by United States Department of Agriculture-National Research Initiative Competitive Grants Program Award number 35301-13314.

## Author contributions

Y.L. performed much of the mapping experiments, the RT-PCR, and most of the pollen mixing experiments. S.A.H. and T.H. performed bioinformatic analysis of the BAC sequence data and gene prediction, and S.A.H. also analyzed the RNA-seq data and genomic sequence data of the knockout mutants. J.L.K. performed some of the genetic tests and performed the screen for the loss-of-function mutants. M.M.S.E. oversaw the project and performed some of the pollen mixing tests and mapping experiments. The manuscript was primarily written by Y.L. and M.M.S.E.

## Additional information

**Competing interests:** M.M.S.E. and J.L.K. are named as inventors on U.S. Patent No. 7,074,984: Cross-incompatibility traits from teosinte and their use in corn (J.L.K., Steven R. Gerrish, and M.M.S.E.). The remaining authors declare no competing interests.

