## [Peer Review File · Nature Communications]

Reviewers' comments:

Reviewer #1 (Remarks to the Author):

Little is known about the molecular mechanisms of self-incompatibility in the grasses. In maize, it was reported decades ago that a couple of genetic factors including Gametophyte factor1-s (Ga1-s) and Ga2-s as well as Teosinte crossing barrier1-s (Tcb1-s) confer unilateral cross-incompatibility (UCI). Two recent publications in Nat. Comm. (Zhang et al.) and Front. Plant Sci. (Lauter et al.) reported the identity of Ga1-s as a pollen-expressed pectin methylesterase (PME) gene named ZmGa1P interacting with ZmPME10 and of a silk-expressed female PME gene named as ZmPME3. While Ga1 and Ga2 are maize factors, Tcb1-s was first reported in teosinte conferring UCI. Tcb1-s pistils block fertilization by maize (tcb1 type) pollen by restricting pollen tube growth. The Tcb1-s locus contains both, tightly linked male (Tcb1-m) and female (Tcb1-f) UCI functions.

By using a complicated genetic approach including mixed pollinations combined with RNA-seq, Lu et al. now report here the identity of Tcb1-f as a PME highly expressed in Tcb1-s silks, but not in silks of standard maize. The gene is almost identical to ZmPME3 mentioned above with only 15 out of 1296 nucleotides being changed. Authors renamed their gene *Pertunda* after a roman god, which is very strange taking into consideration that the genetic factor was named for many years as Tcb1 and they know it encodes a PME – PME3 to be precise. Based on sequence identity authors further conclude that PME3 is involved in cell wall modification. Their data also indicates that modern maize lacking crossing marries have lost Tcb1-s during domestication and breeding.

Although Lu et al. ultimately identified another silk-expressed PME leading to UCI, this finding could not be expected and further highlights the important role of these enzymes for speciation processes. However, although the report has potential, I think it is far from being acceptable for publication in a journal like Nat. Comm.

The report lacks:

- (1.) biochemical data about PME3 activity including approaches to study inhibition of tcb1 pollen using recombinant enzymes;
- (2.) localization studies of the protein;
- (3.) efforts to transfer Tcb1-f to standard maize to establish a UCI barrier.

Reviewer #2 (Remarks to the Author):

The paper by Lu et al. describes experiments to identify the gene underlying the pistil barrier function at the Tcb1-s unilateral incompatibility locus of teosinte. The authors use a combination of map based cloning and RNAseq to identify a pectin methyl esterase gene (PME3, aka *Pertunda*) as the Tcb1-s pistil factor. Crossing experiments show that the PME3 allele from teosinte acts in dominant fashion to selectively eliminate pollen lacking the corresponding male factor (tcb1-m or tcb1). Expression level of PME3 in pistils is correlated with the degree of selection against tcb1 pollen. PME3 variants in maize germplasm match the pistil phenotype (i.e. degree of tcb1 pollen elimination). The results are significant in that they help clarify the mechanism of pollen elimination in wide crosses of maize, and suggest that Tcb1-s (and other UI loci in maize) could play a significant role in preventing gene flow from maize into teosinte. The findings are most relevant to maize domestication, but may also pertain to interspecific reproductive barriers between other crops and their wild relatives.

In general, the paper is well written and the claims are mostly well supported by the results. The paper could be improved in one or two places by adding explanatory details for the non-maize specialist, as detailed below.

My only significant criticism of the paper is that it presents little or no raw data on pollen transmission rates. Figs 1 and 1S show photos of one ear from each cross -- sufficient to show there are qualitative differences in strength of *tcb1* pollen elimination. However it's also obvious from the photos that the expected mendelian ratio of 1:1 in neutral crosses (i.e. on silks of *tcb1* or *Tcb1-m*) was not obtained. (See for example Fig S1-B). This could be either because the pollen mixes were not 1:1, or because the pollen genotypes did not germinate at the same rate, or because the pistil barrier was not completely absent in neutral pistils.

I would like to see a Table (supplementary is fine) showing the actual raw data, including the # plants tested for each cross, the # of kernels with each phenotype (e.g. R1 vs r1) and/or the inferred pollen genotypes (e.g. *Tcb1-s* vs *tcb1*), with the Chi-square goodness-of-fit statistic for conformity to the 1:1 ratio. In crosses that do not fit expected ratios, then data from different crosses should be compared tested for significant differences using the chi-square test of independence. Also, some discussion of possible reasons for non-mendelian segregation would be helpful.

Minor points:

L26-27 I agree that this paper is a major advance in understanding mechanisms of pollen rejection, however identifying the pistil factor encoded by *Tcb1* does not in itself show that this barrier gene has played a role in reproductive isolation in natural populations. This phrase should be reworded accordingly. If the authors do feel that this gene has been shown to play a role in population level processes, then perhaps the introductory text could be modified to highlight the evidence more strongly.

L74 The phrase "any ears without a crossing barrier" could be more explicit (i.e. any ears that don't show significant elimination of *tcb1* pollen compared to the neutral cross, or something along those lines)

L119 "lies IN"

L122 Please explain for the non-maize person the link between the Hopscotch retrotransposon and the Mu transposable element introduced in the cross.

L143 Should 'loss of function' be italicized?

L187 delete 'and'

L206 The term 'incongruity' does not necessarily infer to systems of pollen rejection in which pollen fail to express resistance to a particular barrier gene. This may be the case in maize, but in other plant systems this is not the case, as far as I'm aware. For instance, in Solanaceae self-incompatibility, self pollen are rejected because the lack the SLF function needed to recognize the S-RNase encoded by the same haplotype, yet this is not referred to as 'incongruity'.

Fig. 1B Why does the *tcb1-f* KO line show preferential transmission of R1 pollen? (i.e. more purple than yellow kernels). Shouldn't it be 1:1.

Fig. S1-B Same question regarding transmission rates. Also the ears don't line up with the test crosses in S1-A.

Reviewer #3 (Remarks to the Author):

This is an exciting report of the identification of the Tcb1-s female factor, involving detailed and elegant genetics that I know took many years and a lot of effort. This is difficult work to perform. It's very interesting that the two main gametophyte factor systems described in Zea so far are based on polymethyltransferase enzymes. The demonstration here that several of the loss of function mutants in the Tcb1-female factor have no mutations in the coding region and can revert to function in the presence of mop1 is particularly interesting. My lab has had difficulty characterizing the genetics of some sources of Ga1-s because they appear to be unstable, and it's possible that they could be epi-alleles like these Tcb1 variants (and now we know there are multiple related pseudogenes that could be involved in complex epigenetic silencing processes).

I have almost nothing to criticize in the paper, but a couple of small points:

1. W22 has a very low expression of Pertunda, and this corresponds to the tcb1 allele (line 93). The authors state that Pertunda is not present in the B73 genome (line 99), the closest homologs being pseudogenes at ga1 locus. Then the question is: does W22 have Pertunda? Or is the reported expression of Pertunda an artefact due to short RNAseq reads encoded by some related gene or pseudogene being incorrectly attributed to Pertunda? I BLASTed the Pertunda sequence against both B73 and W22 and the best match appears to be the same pseudogene with 82% homology. It's possible that Pertunda exists in W22 but assembly problems have caused it to be left out, but I wonder if there is some problem assigning RNAseq reads incorrectly to Pertunda.

2. Line 165: 'an ancient Maiz Dulce variety, Jalisco 78 line 1222-2, grown at intermediate altitudes in Southwestern Mexico carries Tcb1-s...': Jalisco 78 is a collection of Maiz Dulce from that region, but note that the 'line' 1222-2 was derived from crossing Jalisco 78 and a temperate commercial line (PHZ51), followed by some selfing. The line itself is not grown in Mexico as a variety.

-Jim Holland

Response to reviews for manuscript NCOMMS-19-03856-T

Explanation of some general changes:

In accordance with Maize Genetics Executive committee requests to reduce the number of names per gene, we have removed the name Pertunda from the manuscript and have used the name, Tcb1-female, throughout as that is already in the literature.

Also, it came to my attention that an additional author, Thomas Hartwig, needed to be included because of his contribution to BAC sequence assembly and gene model prediction.

I also realized that the DNA sequence of the gene was not included anywhere in the manuscript and it is not in supplemental figure 1 (from start to end of transcript model with intron included).

Regarding availability of data:

We have created files for the source data for all of the graphs in the figures (RT-PCR data and transmission data in competition experiments (as also requested by reviewer 2)).

The RNA-seq and genomic DNA-seq data included in the paper are now included in projects at NCBI and those project numbers are included in the manuscript.

Error bars and description of statistical tests are included now.

Response to Reviewer 1:

Points 1 and 2)

While the biochemical and protein localization studies are of course important, we consider them to part of the next study and beyond the scope of this manuscript.

Point 3)

We have stated that the Tcb1 barrier has already been introgressed into maize to establish a UCI barrier, and in fact, these lines are what have been used in the study for ease of crossing.

Response to Reviewer 2:

Regarding how the pollen mix tests on the "neutral" tcb1 ears do not typically achieve a 1 : 1 ratio. We more explicitly state how achieving a 1 : 1 ratio in this mixes is nearly impossible because tassels of different plants cannot be counted on to release the same amount of pollen. In fact, this is why the test to the neutral ear is necessary. There is, however, a slight bias against Tcb1-s pollen on tcb1 silks (as revealed by segregation distortion of a linked genetic marker) and this is also mentioned in the text now with appropriate citation. Also, as requested the raw data for all of these pollen mixes in the graph of figure 3B have been included as a source data file.

The individual comments in the text by reviewer 2:

Minor points:

L26-27 I agree that this paper is a major advance in understanding mechanisms of pollen rejection, however identifying the pistil factor encoded by *Tcb1* does not in itself show that this barrier gene has played a role in reproductive isolation in natural populations. This phrase should be reworded accordingly. If the authors do feel that this gene has been shown to play a role in population level processes, then perhaps the introductory text could be modified to highlight the evidence more strongly.

The language in this statement has been modified to soften this claim now stating that it provides “evidence for a mechanism for reproductive isolation...”

L74 The phrase "any ears without a crossing barrier" could be more explicit (i.e. any ears that don't show significant elimination of *tcb1* pollen compared to the neutral cross, or something along those lines)

The phrase has been changed to “any ears without a crossing barrier that fail to exclude *tcb1* pollen”

L119 "lies IN"

added

L122 Please explain for the non-maize person the link between the Hopscotch retrotransposon and the Mu transposable element introduced in the cross.

Hopscotch is not related to Mu elements but it is common for mutations in these Mu active lines to arise by other mechanism. This is now stated in the text with a reference for an early discovery of this phenomenon.

L143 Should 'loss of function' be italicized?

Italics has been removed here.

L187 delete 'and'

done

L206 The term 'incongruity' does not necessarily infer to systems of pollen rejection in which pollen fail to express resistance to a particular barrier gene. This may be the case in maize, but in other plant systems this is not the case, as far as I'm aware. For instance, in Solanaceae self-incompatibility, self pollen are

rejected because the lack the SLF function needed to recognize the S-RNase encoded by the same haplotype, yet this is not referred to as 'incongruity'.

Incongruity has been removed here and the result of compatibility (rather than rejection) of heterozygous *Tcb1-s/tcb1* pollen by *Tcb1-s* silks is instead cited directly without interpretation.

Fig. 1B Why does the *tcb1-f* KO line show preferential transmission of R1 pollen? (i.e. more purple than yellow kernels). Shouldn't it be 1:1.

The reason for this has been described in the text.

Fig. S1-B Same question regarding transmission rates. Also the ears don't line up with the test crosses in S1-A.

The reason for this has been described in the text, as above. The figure has been modified to align the ears with the relevant test crosses.

Response to Reviewer 3:

Comment 1:

The *Tcb1*-female gene is indeed also absent from W22, which like B73 is functionally *tcb1*. Like B73, W22 also has many copies of pseudogenes for similar sequence at the *ga1* locus. It is stated in the text now that we favor a model that low level expression from these copies are likely the reason for the low detected expression in *tcb1* lines.

Comment 1:

The line 1222-2 is now described more accurately as a derivative from the Maiz Dulce Jalisco78 variety rather than being that variety itself.